# Consequences of Increases in Ambient Temperature and Effect of Climate Type on Digestibility of Forages by Ruminants: A Meta-Analysis in Relation to Global Warming

**DOI:** 10.3390/ani11010172

**Published:** 2021-01-13

**Authors:** Mehluli Moyo, Ignatius Nsahlai

**Affiliations:** Animal and Poultry Science, School of Agricultural, Earth and Environmental Sciences, University of KwaZulu-Natal, Private Bag X01, Scottsville, Pietermaritzburg 3209, South Africa; mehluli.moyo@live.com

**Keywords:** ambient temperature, forage quality, forage digestibility, global warming, ruminants

## Abstract

**Simple Summary:**

This work assessed how the digestion of feeds by cud-chewing animals (ruminants) is affected by animal and feed factors, ambient temperature (AT), and climatic region. The motive for this study was to simulate how forage quality and digestibility would respond under future climate change scenarios. This work allows for predictions to be made on the possible impacts of AT on the digestibility of feeds (viz. roughages, grains, leaves, stems, fruits, and concentrate formulations) consumed by ruminants. This would help farmers to plan and implement strategies for improving feed quality and to align feeding management to ensure improved growth response of ruminant livestock. Increasing AT reduced digestibility parameters consequent upon higher lignification of plant material. The amount of feed that can be potentially digested in a ruminants’ stomach (potential degradability (PD) were highest for concentrates and mixed diets compared to roughages. Potential degradability was lowest for studies carried out in tropical and arid climates compared to cold and temperate climates. Animals fed on diets classified as browse had similar PD compared to those fed on non-browses. Ensilaged feeds had similar PD compared to non-silages. A 1 °C increase in ambient temperature decreased PD by 0.55%, while the fibre content of feeds was projected to increase by approximately 0.4%.

**Abstract:**

This meta-analysis evaluated the effects of ruminant feeding type, ambient temperature (AT), and climatic region on the rumen digestibility of feeds. A dataset on nylon bag degradability parameters bearing the chemical composition of roughages, grains, leaves, stems, fruits, concentrates and diets given to animals, climate type, and AT were compiled. Data were analysed using mixed model regression and simple linear regression methodologies. Negative correlations between AT and degradability parameters were observed. Potential degradability (PD) and slowly degradable fraction (‘b’) were higher for concentrates and mixed diets compared to roughages. Intermediate feeders had slower rates of degradation (‘c’) compared to grazers. Potential degradability was highest for studies carried out in cold and temperate climates compared to tropical and arid climates. A 1 °C increase in AT decreased PD by 0.39% (roughages), 0.76% (concentrates), and 2.41% (mixed diets), with an overall decrease of 0.55% for all feed types. The “b” fraction decreased by 0.1% (roughages), 1.1% (concentrates), 2.27% (mixed diets), and 0.35% (all feed types) for every 1 °C increase in AT. Increasing AT by 1 °C increased the neutral detergent fibre content of feeds by 0.4%. In conclusion, increases in AT increased the neutral detergent fibre content of feeds, lowering PD, “b”, and “c” of dry matter in the rumen.

## 1. Introduction

Microbial degradation is important in determining the digestibility [1,2,3,4] and rate of passage of fibrous feeds in the rumen, growth rate [5], and feed intake of ruminants [3,4,6,7,8]. Numerous studies [9,10] have concluded that the degradability of feeds in the rumen is greatly influenced by feed properties being digested such as neutral detergent fibre and crude protein content.

Equations for predicting degradation parameters mainly use properties of feed being degraded as major prediction variables. It is known that the degradation of roughage diets depends on the composition of microbes (bacteria, protozoa and fungi) in the rumen [9]. Therefore, the potential extent of degradation of roughages in the rumen partly depends on rumen ecology as determined by diets fed to animals and outflow rates of liquid and solid in the rumen [9,10]. The duration for incubating feeds in the rumen in degradation studies vary between temperate and tropical regions, suggesting that potential degradability of feeds may be influenced by climate and ambient temperature. Global temperatures are expected to increase by just over 1 °C per annum, and global warming is projected to reduce forage quality by lowering digestibility and crude protein content of feeds [11,12]. Few studies [13,14,15,16,17] have documented the extent to which a unit increase in ambient temperature would decrease forage digestibility. Studies need to simulate how forage quality would respond under future climate change scenarios. The ability to predict how digestibility would be affected by ambient temperature can be a useful tool in planning and implementing strategies of improving feed quality and feeding management to ensure improved growth response of ruminants.

The magnitude of change in the rumen digestibility parameters of feeds as affected by changes in ambient temperature are not well documented. Despite the large number of research publications on the degradation of feeds in the rumen, few attempts have been made to synthesize a global view of the main determinants of degradation. This study summarized findings across published studies to establish the main determinants of degradation of feeds in the rumen using meta-analysis methodology. The aim of this study was to determine the effects of ambient temperature and climate type on rumen digestibility and chemical composition of forages.

## 2. Materials and Methods

### 2.1. Creation of Dataset

Data were collected from studies that measured degradability parameters which included the soluble fraction (a), slowly degradable fraction (b), potential degradability (PD), lag and rates of degradation (c) of feeds in the rumen. This meta-analysis was carried out from studies that met all of the following criteria: (1) studies were published in peer-reviewed journals, (2) in-sacco degradability was done using the nylon-bag technique, (3) studies reported the degradability of dry matter in the rumen, (4) studies reported the degradability of any feed including roughages, grains, leaves, stems, fruits, concentrates and diet formulations, and (5) studies stated the feeds or diets and any supplementary feeds fed to the animals, (6) animals were fed ad libitum. Observations on the degradability of organic matter and neutral detergent fibre were not included in the dataset. This dataset had observations of degradation parameters from wild and domesticated ruminants. Although publications collected to create the dataset might not be exhaustive of all published literature, studies used were readily accessible and available for use. Data were collected from 111 studies carried out worldwide between 1985 and 2018.

Factors affecting degradability were identified in each of these studies and were categorized into two main groups: (1) diet properties (i.e., fed to the animal) and (2) feed sample properties (i.e., incubated in the rumen), while the effects of ruminant feeding type and environment (climate and management) related factors were also included as shown on Figure 1. Potential degradability (PD) was calculated in studies that did not report it using the formulae: PD = a + b.

### 2.2. Diet Properties

Diet properties were used to account for the effect of rumen ecology on degradation. Dietary nutritional attributes included in this dataset were neutral detergent fibre (NDF), protein-free cell contents (PFCC) which include starch, sugars, vitamins, pectins, fats and minerals, and crude protein (CP) contents of entire diet (all in g/kg). Protein free cell contents (PFCC) of diets fed to animals was calculated using the formula: PFCC = 1000 − (NDF + CP). Discrete dietary properties that affect degradation parameters included in the dataset were diet class and diet subclass. Diet class classified diets either as a silage or non-silage and diet subclass classified diets either as a browse or non-browse. Thus, diet properties included in the dataset were NDF, PFCC, and CP contents of entire diet, diet class, and diet subclass.

### 2.3. Feed Sample Properties

Feed sample properties that affect degradability were identified from literature including Bach et al. [9] and Moyo et al. [10]. Feed sample properties were particle size (mm), feed class, feed subclass, and feed compositional attributes (DM, dry matter; CP, crude protein; NDF, neutral detergent fibre, ADF, acid detergent fibre; HEM, hemicellulose, and ash all in g/kg). Particle size of degradation samples (mm) was determined from the screen size used to grind each sample for incubation. Hemicellulose (HEM) content was calculated in studies that did not report it using the formulae: HEM = NDF − ADF. Discrete feed factors that affect degradation parameters included in the dataset were: feed class, feed subclass and feed type. Feed class classified feeds either as a silage (=1) or non-silage (=0); feed subclass classified feeds either as urea-treated (=1) or untreated (=0); and feed type classified feeds either as a roughage (any part of a plant that is not a fruit or grain = 1), concentrate (parts of a plant that constitutes a grain or fruit = 2) or mixed diet (mixture of roughages and concentrates = 3). Thus, feed properties included in the dataset were DM, CP, NDF, ADF, HEM, particle size of feed, feed class, feed subclass, and feed type.

### 2.4. Ruminant Species and Climate

To account for the effect of ruminant species feeding type on degradation, ruminants were separated into 3 main feeding types according to the classification by Hofmann [18] as: grazers or roughage selectors (buffalo, cattle and sheep) and intermediate feeders (goats). Effects of climate on degradation were accounted for by identifying the location where each study was done and classifying the climate of each study site using the updated Köppen-Geiger climate classification system according to Peel et al. [19]. Studies fell into 15 climatic regions namely; tropical rain forest (Af), tropical savannah climate (Aw), hot arid desert climate (BWh), hot arid steppe climate (BSh), cold arid steppe climate (BSk), dry temperate climate with hot summers (Csa), dry temperate climate with warm summers (Csb), dry winter temperate climate with hot summer (Cwa), dry winter temperate climate with warm summers (Cwb), hot summer temperate climate without dry season (Cfa), warm summer temperate climate without dry season (Cfb), cold dry climate with warm summers (Dsb), cold climate with hot summers and no dry season (Dfa), cold climate with warm summers and no dry season (Dfb), and cold climate with warm summers and no dry season (Dfc). The effect of climate type was done by allocating each climatic region into either tropical (A), arid (B), temperate (C) and cold (D) climate types according to the Köppen–Geiger climate classification system. Countries where and years when each study was done were obtained. Data on ambient temperature for each of these regions where the studies were done was obtained from Harris et al. [20].

### 2.5. Data Analysis

Data were normalized to meet assumptions of homogeneity of variance using the logarithmic transformation. A meta-analysis was done using the mixed model regression procedure of SAS 9.3 software (SAS Institute Inc., Cary, NC, USA) according to St-Pierre [21] and Sauvant et al. [22] to determine the main effects of feed type, ruminant type, ruminant feeding type, and climate on rumen degradation of feeds. A model with discrete predictor variables (feed type, ruminant type, ruminant feeding type and climate) considered as fixed effects were used. The fixed effects of dry matter, ash, and crude protein contents of feed samples, particle size of feed samples, whether feed samples were silage or non-silage, treated with urea or untreated, crude protein, and PFCC contents of diets fed to animals, whether the diet was a silage or non-silage, browse or non-browse, whether animals were fed indoors or grazing outdoors, and ambient temperature were considered as covariates. Study × incubation time interactions from different studies were considered as random effects. Data were weighted by the number of animals in each study and the standard errors of the mean [22].

A second model with discrete predictor variables (feed type and climate) considered as fixed effects were used to determine the main effects of feed type and climate on chemical composition of feeds/diets incubated for rumen degradation. The fixed effects of whether or not feed samples were a silage or non-silage and whether or not feed samples were treated with urea or untreated and ambient temperature were considered as covariates. Study × incubation time interactions from different studies were considered as random effects. Data were weighted by the number of animals in each study. Least square means were used to compare differences among means in the case of discrete predictor variable. The probability threshold for significance of fixed and random study effects for meta-analyses were considered at *p* < 0.05. The correlation procedure was used to establish the Pearson correlation coefficients of any two input predictor variables.

## 3. Results

In some studies, not all the variables of interest were reported, therefore, the number of observations across variables was not uniform (Table 1). There were large differences between minimum and maximum values in the dataset for degradability parameters, proximate composition of diets (DCP, DNDF and PFCC) fed to animals and feed samples (DM, CP, NDF, ADF, HEM, and ash) degraded in the rumen. The variability in ambient temperatures (CV = 45.66%) of regions where these studies were done and incubation times of feeds in the rumen among studies was high.

There were significant positive correlations between DCP and PD (r < 0.15; *p* < 0.05), and length of incubation and PD (r < 0.15; *p* < 0.05) (Table 2). Significant negative correlations (*p* < 0.05) between AT and degradability parameters (a, b, c, and PD) were observed. Correlations between CP and degradability parameters (a, b, c, and PD) were positive and significant. There were significant negative correlations between ADF and degradability parameters (a, b, c, and PD).

Concentrates had the highest solubility compared to roughages and mixed diets, while the potential degradability and the slowly degradable fraction were higher for concentrates and mixed diets compared to roughages (Table 3). Ruminants classified as grazers had faster rates of degradation compared to intermediate feeders. Ruminant type had no effect on all degradation parameters but the rate of degradation of feeds in the rumen. The rate of degradation in small ruminants (goats and sheep) was lower than in large ruminants (buffalo and cattle). The PD was highest for studies carried out in cold and temperate climates compared to tropical and arid climates.

Rumen ecology as influenced by diet properties fed to animals affected the rate of degradation and PD of feeds in the rumen (Table 4). Animals fed on diets classified as browse had similar PD (466 ± 168.6 g/kg) with those fed on non-browse (671.9 ± 64.93 g/kg) diets. Silage diets had similar PD (609 ± 105.04 g/kg) to non-silage (529 ± 94.62 g/kg) diets. The solubility of feeds was higher for animals fed non-silage diets (224.7 ± 181.49 g/kg) compared to those fed on silage diets (6.3 ± 191.65 g/kg).

Relationships between PD and AT were more linear (*p* < 0.0001) than they were quadratic (*p* = 0.0137). Significant negative linear relationships were observed between ambient temperature (AT) and the slowly degradable fraction of fibre (b) and potentially degradability (PD) as shown on Table 5. A 1 °C increase in AT decreased PD by 0.39% (roughages), 0.76% (concentrates) and 2.41% (mixed diets). The “b”-fraction decreased by 0.1% (roughages), 1.1% (concentrates), and 2.27% (mixed diets) for every 1 °C increase in AT. Regression equations demonstrated that the PD and b decreased by approximately 0.55% and 0.35% for every 1 °C increase in AT. A significant positive linear relationship was observed between PD and dietary crude protein (DCP). Large differences were found for slopes of regression equations among all three feed types. Negative effects of increasing AT on PD and b were more pronounced in mixed diets, followed by concentrates and less on roughages. Regression analysis showed that a unit increase in DCP content improved PD of mixed diets six times more than roughages. The PD increased slightly by approximately 0.05% for every unit increase in DCP content for all feed samples.

Significant positive linear relationships were observed between ambient temperature (AT), the dry matter (DM), and neutral detergent fibre contents of feeds, as shown in Table 6. Regression equations demonstrated that DM gradually increased by 7% (mixed diets) and 0.4% (roughages) for every 1 °C increase in AT. Increasing ambient temperature by 1 °C increased neutral detergent fibre content of feeds by 0.4%. Test of slopes showed DM content of mixed diets increased by 15 times more compared to roughages for every 1 °C increase in AT. The rates of degradation calculated using the no time-lag tended to be higher than the rates of degradation estimated from the model that accounts for time lag (Table 7). The Lag time was longer for roughages compared to concentrates. Ruminant type and feeding type affected estimation of rates of degradation using the no time-lag model. Roughages had low CP and high NDF contents compared to concentrates and mixed diets (Table 8).

## 4. Discussion

Changing climatic conditions towards global warming are projected to reduce forage quality, but little is known concerning the extent of reduction in a parameter such as the digestibility of feeds. Determining the extent of the overall effects of climates and global warming on feed nutritional composition and in-vivo digestibility using controlled experiments is challenging because of the need for replication of a wide range of ambient temperature treatments and other environmental factors. A meta-analysis evaluation would help us to infer on the effect of ambient temperature on digestibility and how forages (feeds) would respond under future climate change scenarios. The main motive for this study was to simulate how forage quality and digestibility would respond under future climate change scenarios. It is worth noting that, in the estimation of degradation parameters, different mathematical models give rise to a variation in these estimates and discrepancies are highlighted in various sections of this discussion.

### 4.1. Implications of Using the no Time-Lag and Time-Lag Models on the Rate of Degradation

Degradability parameters (a, b, c and PD) are generally predicted by fitting dry matter loss from nylon bags using 2 types of models, one that accounts for and another that does not account for the time-lag. The no lag model by Orskov & McDonald [23] takes the form Y = a + b (1 − e^−ct^) and is suitable for feeds with low fiber content; and the model that accounts for the time-lag by McDonald [24] takes the form Y = a + b (1 − e^−c(t − L)^) and is suitable for fibrous feeds; where: Y = degradability at time (t), a = intercept (rapidly soluble fraction or solubility), b = slowly degradable fraction, c = rate of degradation of the slowly degradable fraction (b) and L = lag time. The time-lag is the period of colonization occurring between the washing away of solubles and the initial commencement of fermentation of feed by bacteria. Quantification of this time lag is crucial in determining the exact rate of degradation of a feed particle in the rumen. Not accounting for time-lag can either depress (for roughages) or inflate (for all feed types) the rate of degradation and gives erroneous estimates because there is time required for feed particles to be colonized before degradation commences. Hence, the model that does not account for time-lag can underestimate the rate of degradation of the slowly degradable fraction. For reporting purposes, workers should consider predicting degradation parameters using the model that accounts for the time-lag.

An option for workers that prefer using the model that does not account for time-lag, an additional parameter called the “wash-loss” should be reported instead for example Navatne & Ibrahim [25] and Umunna et al. [26]. The wash-loss value can be used to re-calculate the rate of degradation. The sigmoid shaped degradability curve (using the no-time lag model) can be reconstructed and plotted together with the linear wash-loss curve. The two curves intersect at the coordinates (Lag time; wash-loss value). The lag time for degradation to occur can be determined from the points of intersection of these two curves. The rates of degradation and time lag preceding degradation in studies that use the no time lag model can then be recalculated. The new rates of degradation (‘c’) can be calculated using the time lag model at the point of inflection where the rate change was fastest normally assumed to occur at an approximate range of 9–24 h of incubation for most feed types.

Computation of the rates of degradation using the time-lag model may result in negative values for time lag being reported and these negative lag times have no biological meaning. It is more appropriate to assume all negative lag-times for degradation to commence at 0 h, especially feeds classified as concentrates. Studies that seek to predict or simulate degradation rates should separate datasets based on which of the two models were used to estimate degradation rates to avoid under predicting the rate of degradation, in a similar way to the meta-analysis of Busanello et al. [27].

### 4.2. Effects of Climate and Ambient Temperature on Degradation of Forages

Expectedly, the potential degradability and the rate of degradation of roughages were lower than that of concentrates, while feeds from cold and temperate climates were digested faster than feeds from tropical and arid climates. The effect of climatic region on digestibility of roughages was evident from a study by Nsahlai & Apaloo [28]. In their evaluation of temperate roughage-based digestibility models, Illius & Gordon [29] predicted the digestibility of tropical roughages, whereby the overall trend between the observed and the predicted digestibility was positive, achieving accuracies of 36–52%. Nsahlai & Apaloo′s [28] evaluations using tropical roughages did not compare well with accuracies of approximately 70% obtained from model evaluation using temperate roughages. These low levels of accuracy of simulating the digestibility of low-quality roughages commonly grazed and fed to ruminants in the tropics may have been due to the effect of ambient temperature on digestibility of plant material [28], for which the effect has been indexed. The effects of AT on degradability of feeds in the rumen may occur in two ways: firstly, through changing the chemical composition of feed sample and secondly by possibly altering rumen physiological processes.

There are suggestions that increases in AT would affect the degradability of feeds in the rumen by increasing the lignin and NDF [30] and decreasing the CP content of feeds, thereby lowering the rate of degradation and PD of feeds [11]. Elevated levels of AT increase NDF, but decrease the CP content of feeds, lowering feed quality [12]. In this study, gradual increases in ambient temperate had a negative linear effect on the PD and b-fraction. This observation can be partially supported by the positive relationship between NDF and AT, which increases NDF content of feeds with increases in AT, lowering PD of feeds. Miaron & Christopherson [31] observed a quadratic relationship between apparent digestibility (Y) and temperature (X) that took the form Y = 69 − 0.188x − 0.017x^2^. Findings from this meta-analysis suggest that, with global warming, the quality of feeds, based on rumen degradability would most likely decrease by 0.6% for every 1 °C increase in ambient temperature. A favourable increase in rumen degradability, PD, would be expected in regions where temperatures are predicted to decrease due to climate change.

Contrary to trends observed in this study, a decrease in AT from 21 °C to 0 °C did not significantly affect nylon bag degradability of cell wall constituents, although feed form × ambient temperature interactions affected the rate of degradation of cell wall constituents [32]. Again, prolonged exposure of sheep and steers to cold temperature of approximately 2 to 5 °C would cause a depression in apparent dry matter digestibility of 0.2% and 0.08% per degree Celsius compared to sheep and steers exposed to temperatures of 22 to 25 °C, respectively. The decrease in digestibility at low ambient temperatures can be attributed to increases in the rate of passage of digesta in through the rumen [13], limiting time taken for fermentation to occur. Apparent digestibility of dry and organic matter on average were 17% higher at 28 °C than at 10 °C in steers [31]. The regression of pooled data from 16 studies showed a positive trend between digestibility and ambient temperature [17], contrary to the trend from this study where rumen digestibility decreased with an increase in AT.

Kennedy et al. [14] reported a decrease in digestibility of organic matter in the rumen (F) with exposure of sheep to AT of −1 to 1 °C and 18 to 21 °C, and was highly correlated to solid digesta passage rate (kp) in the rumen (F = 14.57 kp + 239; R^2^ = 0.90, SE = 32.6). Effects of AT on digesta passage rates are equivocal [33]. Low AT (−1 to 1 °C) caused faster liquid and solid digesta passage rates in sheep compared to high AT (18–21 °C) [14,15,16], but did not have an effect on digesta passage rates [32], while high AT (41 °C) caused faster liquid passage rates compared to low AT (26 °C) in swamp buffalo [34]. Theoretically, degradability of DM in the rumen is expected to decrease with an increase in AT, due principally to deceased rate of digestion consequent upon higher lignification and faster rates of passage of digesta in the rumen at low AT. Fast passage rates of digesta in the rumen decreases the maintenance energy requirements and mean age of microbial population causing an abundance of young microbial cells with high growth potential which is lacking in old bacterial cells [35]. Abundance of young bacterial cells in the rumen translates to increased rate of degradation and high PD in the rumen. Findings of this study that showed a linear decrease in PD with increasing AT are supported by this theory. Another point of contention would be that an increase in passage rate of solid digesta in the rumen would reduce mean retention time of feed in the rumen for microbial fermentation. Low mean retention times as a result of sheep exposure to low AT (−1 to 1 °C) would be expected to reduce digestibility of feed in the rumen compared to sheep exposed to high AT (18 to 21 °C), consistent with findings of Kennedy et al. [14,15] and Kennedy & Mulligan [16]. Empirical findings available on the nature of the relationship between AT and digestion gave a different trend to those obtained from this study. These may be attributed to that most published studies have evaluated relatively narrow ranges of ambient temperatures and at the very best compared two or three temperature treatments [17,31]. Increases in AT have an overall effect of increasing lignin and ultimately NDF content of feeds [11,30]. The overall positive linear trend between NDF content of feeds and AT, and a significant negative correlation between NDF content of feeds and PD observed in this study, strongly support the theory that increases in AT would most likely cause a decrease in PD of feeds in the rumen.

Predictions from this study showed a sharp decrease in PD of concentrates compared to roughages. The rate of decrease of degradability of the slowly degradable fraction per unit increase in AT followed the trend: mixed diets > concentrates > roughages. Roughages had the least negative response in PD to increases in AT. High NDF and ADF contents of feeds reduce dry matter digestibility [36]. Because roughages had high NDF content compared to concentrates, it was expected that concentrates would have a much greater rate of decrease in PD per unit increase of AT. The digestibility of concentrates was more susceptible to influences of ambient temperature, contrary to Christopherson & Kennedy [17] where digestibility of slowly degradable forages appeared to be more susceptible to influence by ambient temperature induced changes compared to rapidly degradable forages. The higher the ambient temperature, the lower the CP of feeds, as evidenced by the significant negative correlation between these two variables. This would most likely decrease the PD of feeds in the rumen.

Trends observed in this study suggested that average crude protein content of feeds incubated in the rumen was highest for cold climates, followed by temperate climate, then arid climates and lowest for tropical climates. Feeds from arid desert climates characterised by low erratic rainfall had similar crude protein content to feeds in high rain fall tropical rain forest climates. This may be because plants from arid desert climates grow fast and reach maturity quickly when water is available and deposit less lignin making the resultant feeds to be of good quality with relatively high CP and low NDF. The potential degradability of feeds in cold climates (Dfa and Dfb) was lowest compared to tropical, arid and temperate climates, although feeds from cold climates had one of the highest crude protein contents.

### 4.3. Effects of Diet and Feed Sample Chemical Content on Degradation

There was a significant positive relationship between PD and dietary crude protein. Increasing crude protein content of diets fed to ruminants increased PD of feeds in the rumen. Riaz et al. [36] also reported a positive influence of dietary crude protein on dry matter digestibility in buffaloes, cattle, sheep and goats. Bonsi et al. [37] showed that graded levels of *Sesbania sesban*, which were used to gradually increase dietary crude protein content, tended to increase the rate of degradation. A constant supply of energy and crude protein from the diet is required for bacterial population growth and proliferation responsible for most degradation in the rumen. Thus, an increase in dietary crude protein is expected to increase the PD of feeds. This trend is substantiated by the significant positive correlation between dietary crude protein content and degradation parameters (a, b, and PD) observed in this study. The response of degradation of mixed diets to increased dietary crude protein levels was higher than for roughages. This is because roughages generally tend to be of lower quality (high NDF and low crude protein contents) than mixed diets, lowering the response of rate of increase of degradation of roughages to incremental levels of dietary crude protein.

The PD for roughages was lower than that of concentrates and mixed diets. Due to better proximate nutritional composition, it is expected that the digestion of concentrates would be higher than that of roughages. The faster rates of digestion and high digestibility of concentrates (grain meals, seeds and fruits) compared to roughages [38], concentrates are incubated for much shorter periods compared roughages. The high PD of concentrates compared to roughages may be linked to lower average duration of incubation times observed in studies that measured degradation of concentrates (48–70 h) compared to roughages (117 h) and mixed diets (275 h). The average incubation time of feeds in all studies in the dataset was approximately 120 h, showing that concentrates reach their PD relatively earlier (48 h). Potential degradability is a feed property that is affected by the rumen ecology because of degradation rate and length of incubation of fibrous feeds [39], where shorter durations of incubations point to imprecise estimates. A shorter incubation time of feeds in the rumen can bring about erroneous estimates of PD by either depressing the PD for roughages or inflating the PD when the degradation curve is terminated at a point where the curve is still rising before it reaches a horizontal asymptote. Although not tested in their study, findings of Tolera & Sundstøl [40] showed increased dry matter disappearance (DMD) with increasing incubation time for maize stover (DMD4h = 18.5%, DMD24h = 36%, DMD48h = 51%, and DMD96 h = 66%) and *Desmodium intortum* hay (DMD4h = 29.1%, DMD24h = 54.4%, DMD48h = 65.1%, and DMD96h = 67.2%). This trend is consistent with the positive significant correlation between incubation time and PD observed in this study. A meta-analysis finding by Busanello et al. [27] showed that the degradation of dry matter was similar for meals (oil cakes) and grains (concentrates).

Climatic region did not affect rates of degradation of feeds in the rumen. This may have been due to the similar neutral detergent fibre content of feeds from all climatic regions. Feeds of high ash content had better PD in the rumen, probably facilitated by the catalytic effects of ash on bacteria in the rumen. Contrary to our findings, ensilage of feeds tended to increase the effective degradation of dry matter and acid detergent fibre in the rumen compared to fresh feed samples [41].

The negative effects of ambient temperature on forage quality have major implications to small ruminant feeding. Small ruminants possess a small rumen fermentation capacity with respect to their high metabolic requirements and, consequently, select and consume a better-quality diet, which is retained and digested for short periods rendering reduced potential for maximal degradation of low-quality roughages [42,43]. Under future climate change scenarios where increases in AT are anticipated, the observed trend in the decrease in digestibility of feeds with increases in AT in this study cannot be overlooked. Although the 0.6% decrease in PD per 1 °C increase in AT (−5.9 to 28.2 °C) may seem small, it may have dire consequences to ruminant livestock performance. The predicted decrease in PD would be most severe in tropical areas where most grass species are generally of low quality compared to temperate grasses. Findings from this study predicted a sharp decrease in PD of concentrates (−0.7%), in which concentrates had the sharpest decline compared to roughages and mixed diets. Cereal grain concentrates are mainly used to supplement ruminant livestock in tropical areas, where ambient temperatures are generally high, suggesting that most cereal grains will decrease in digestibility. The implications for ruminant livestock would be a decrease in their performance attribute. Alternative feeding strategies, such as urea supplementation and treatment of poor-quality roughages would need to be adopted to improve the nutritional status of ruminant livestock [44,45]. The adoption of drought-tolerant ruminant livestock species and/or breeds that are capable of efficiently utilising poor quality roughages needs to be undertaken. This would entail exploiting local or indigenous breeds of cattle, sheep, and goats.

## 5. Conclusions

Increases in ambient temperature increased the neutral detergent fibre content of feeds, lowering the potential degradability of dry matter in the rumen. A 1 °C increase in AT decreased PD by 0.39% (for roughages), 0.76% (for concentrates), and 2.41% (for mixed diets). The slowly degradable fraction decreased by 0.1% (for roughages), 1.1% (for concentrates), and 2.27% (for mixed diets) for every 1 °C increase in AT. Overall, a 1 °C increase in AT decreased PD and “b” by 0.55% and 0.35%, respectively. Increasing ambient temperature by 1 °C increased neutral detergent fibre content of feeds by 0.4%. The predicted decrease in rumen digestibility of feeds with ambient temperature would be most severe in tropical and arid regions compared to cold and temperate regions. A sharp decrease in the potential degradability of concentrates (−0.7%) was predicted, in which concentrates had the sharpest decline compared to roughages. Findings from this study can be incorporated into the initial mitigating measures aimed at improving the feeding value of poor-quality roughages, especially crop residues such as straws and stovers. The effect ambient temperature on potential degradability in the rumen provides strong evidence of why ambient temperature should be accounted for in models that seek to predict digestibility in the rumen.

## Figures and Tables

**Figure 1 animals-11-00172-f001:**
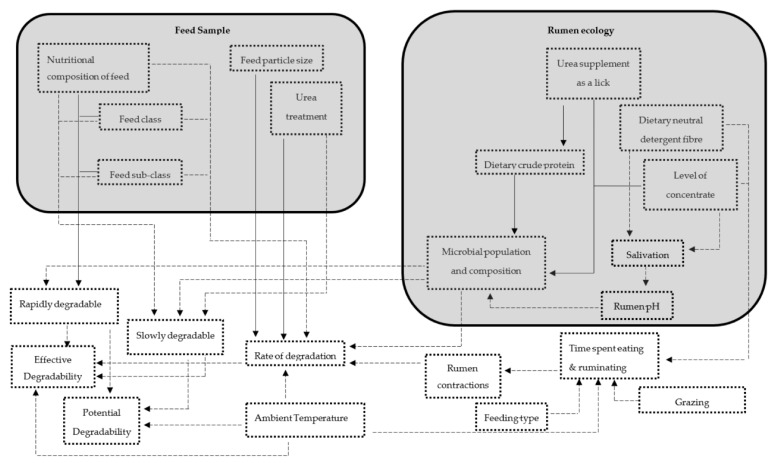
Factors affecting degradation of feeds in the rumen.

**Table 1 animals-11-00172-t001:** Descriptive statistics of diet, feed and climatic factors affecting degradation of feeds in the rumen.

Diet	N	Max	Min	Mean ± SD	SEM	CV (%)
Crude protein (g/kg)	1006	311	20	124 ± 44.5	1.41	35.98
Neutral detergent fibre (g/kg)	1006	913	129	565 ± 124.4	3.92	22.04
Protein-free cell contents (g/kg)	1006	740	48	311 ± 109.5	3.45	35.24
Feed sample						
Dry matter (g/kg)	1015	992	70	713 ± 302.8	9.51	42.47
Crude protein (g/kg)	1009	519	19	119 ± 75.6	2.38	63.67
Neutral detergent fibre (g/kg)	1006	919	69.3	558 ± 193.7	6.14	35.01
Acid detergent fibre (g/kg)	1006	715	29	357 ± 136.2	4.31	38.48
Hemicellulose (g/kg)	1006	524	5.8	202 ± 100.4	3.17	50.04
Ash (g/kg)	1009	330	11	87 ± 39.9	1.25	45.91
Particle size (mm)	1015	100	0.5	3.3 ± 6.72	0.21	206.4
Soluble fraction (g/kg)	945	751	27	214 ± 119.3	3.89	55.81
Slowly degradable fraction (g/kg)	947	984	64	502 ± 149.9	4.87	29.86
Rate of degradation (per h)	997	2.148	0.007	0.050 ± 0.085	0.003	170.5
Potential degradability (g/kg)	974	1000	31	711 ± 151.1	4.84	21.22
Lag (h)	375	17.90	0.00	2.24 ± 2.763	0.143	123.4
Climate						
Ambient temperature (°C)	1015	28.2	−5.9	17.8 ± 8.14	0.26	45.66
Experimental factors						
Incubation time (h)	1015	336	36	117 ± 79.3	2.49	67.58
No. of replicates used	977	12	1	3.7 ± 1.7	-	46.28

CV, coefficient of variation; SEM, standard error of the mean; SD, standard deviation.

**Table 2 animals-11-00172-t002:** Pearson correlations between diet, feed and climatic factors affecting degradation of feeds in the rumen.

	Diet Attributes			Feed Sample Attributes						Environmental Factors		Degradability Parameters				
	DCP	DNDF	PFCC	DM	CP	Ash	NDF	ADF	HEM	AT	IT	a	b	c	PD	Lag
DCP	-	−0.50 ***	0.15 ***	−0.17 ***	0.19 ***	0.02 ^NS^	−0.01 ^NS^	−0.08 *	0.08 **	−0.24 ***	0.26 ***	0.07 *	0.09 **	0.05 ^NS^	0.15 ***	0.15 **
DNDF		-	−0.94 ***	−0.03 ^NS^	−0.11 ***	0.16 ***	0.27 ***	0.22 ***	0.23 ***	0.44 ***	−0.09 **	−0.09 **	−0.04 ^NS^	−0.08 *	−0.11 ***	0.02 ^NS^
PFCC			-	0.10 **	0.05 ^NS^	−0.19 ***	−0.31 ***	−0.22 ***	−0.30 ***	−0.40 ***	−0.01 ^NS^	0.07 *	0.01 ^NS^	0.07 *	0.07 *	−0.07 ^NS^
DM				-	−0.17 ***	−0.24 ***	0.13 ***	0.20 ***	−0.03 ^NS^	0.09 **	−0.27 ***	−0.25 ***	0.04 ^NS^	0.02 ^NS^	−0.16 ***	−0.22 ***
CP					-	0.18 ***	−0.56 ***	−0.50 ***	−0.41 ***	−0.08 **	0.18 ***	0.23 ***	0.07 *	0.26 ***	0.25 ***	−0.10 *
Ash						-	−0.01 ^NS^	0.04 ^NS^	−0.08 *	0.19 ***	0.12 ***	−0.01 ^NS^	0.04 ^NS^	−0.03 ^NS^	−0.02 ^NS^	0.21 ***
NDF							-	0.87 ***	0.75 ***	0.16 ***	0.01 ^NS^	−0.38 ***	−0.01 ^NS^	−0.21 ***	−0.32 ***	0.29 ***
ADF								-	0.33 ***	0.20 ***	−0.02 ^NS^	−0.39 ***	−0.09 **	−0.18 ***	−0.41 ***	0.38 ***
HEM									-	0.04 ^NS^	0.03 ^NS^	−0.21 ***	0.12 ***	−0.15 ***	−0.07 *	0.14 **
AT										-	−0.13 ***	−0.13 ***	−0.19 ***	−0.11 ***	−0.30 ***	0.14 **
IT											-	−0.07 *	0.19 ***	−0.01 ^NS^	0.15 ***	0.23 ***
a												-	−0.43 ***	0.08 *	0.38 ***	−0.02 ^NS^
b													-	−0.08 *	0.67 ***	−0.18 ***
c														-	−0.01 ^NS^	−0.08 ^NS^
PD															-	−0.20 ***
tL																-

DCP, dietary crude protein; DNDF, dietary neutral detergent fibre; DM, dry matter; CP, crude protein, NDF, neutral detergent fibre; ADF, acid detergent fibre; HEM, hemicellulose; PFCC, protein-free cell contents; AT, ambient temperature; IT, incubation time; a, soluble fraction; b, slowly degradable fraction; PD, potential degradability and c, rate of degradation; tL, time lag.* *p* < 0.05; ** *p* < 0.01; *** *p* < 0.001; NS, not significant (*p* > 0.05).

**Table 3 animals-11-00172-t003:** Effects of feed sample and diet properties, ambient temperature, ruminant type, feeding type, and climatic region on rumen degradation of feeds.

Test of Fixed Effects	Degradation Parameter Estimates (Mean ± SE)				
Effect of Feed Type	a (g/kg)	b (g/kg)	c (per h)	PD (g/kg)	Lag (h)
Roughages	212 ± 53.68	491 ± 99.24	0.046 ± 0.0072	697 ± 56.9	2.78 ± 4.311
Concentrates	237 ± 60.57	538 ± 99.84	0.080 ± 0.0088	780 ± 60.4	0.96 ± 4.411
Mixed diets	199 ± 103.76	634 ± 190.80	0.051 ± 0.0154	833 ± 103.2	0.00 ± 6.62
Significance	***	***	*	**	NS
Effect of climatic region					
Tropical climates					
Af	92 ± 326.81	619 ± 731.79	0.029 ± 0.0281	643 ± 128.8	6.07 ± 6.098
Aw	200 ± 90.55	466 ± 195.93	0.039 ± 0.0161	666 ± 87.1	1.54 ± 5.57
Arid climates					
BSh	176 ± 96.07	404 ± 201.34	0.037 ± 0.0180	581 ± 102.2	-
BSk	160 ± 95.27	537 ± 196.73	0.068 ± 0.0210	697 ± 93.7	5.13 ± 5.593
BWh	151 ± 484.22	358 ± 718.25	0.052 ± 0.0444	606 ± 203.4	-
Temperate climates					
Cfa	266 ± 92.72	476 ± 223.75	0.051 ± 0.0154	697 ± 87.5	0.62 ± 5.253
Cfb	258 ± 70.82	535 ± 162.21	0.052 ± 0.0143	792 ± 77.9	2.79 ± 6.03
Csa	288 ± 89.65	437 ± 176.15	0.055 ± 0.0189	725 ± 94.1	2.60 ± 5.723
Csb	231 ± 135.18	491 ± 274.70	0.056 ± 0.0319	722 ± 121.6	-
Cwa	247 ± 103.03	476 ± 240.21	0.064 ± 0.0207	722 ± 117.8	0.58 ± 5.828
Cwb	185 ± 82.56	533 ± 159.05	0.036 ± 0.0134	715 ± 77.7	2.71 ± 5.60
Cold climates					
Dfa	241 ± 183.70	447 ± 266.03	0.080 ± 0.0271	688 ± 112.7	1.31 ± 5.956
Dfb	120 ± 141.29	615 ± 478.79	0.234 ± 0.0511	735 ± 115.0	2.03 ± 5.840
Dfc	98 ± 333.95	624 ± 609.91	0.027 ± 0.0426	722 ± 210.8	-
Dsb	213 ± 100.76	478 ± 233.82	0.032 ± 0.0274	691 ± 150.8	-
Significance	NS	NS	NS	NS	***
Effect of climate type					
Tropical	193 ± 63.90	476 ± 147.93	0.038 ± 0.0113	664 ± 54.105	2.62 ± 1.828
Arid	166 ± 55.52	439 ± 133.61	0.050 ± 0.0128	621 ± 55.482	3.14 ± 3.200
Temperate	237 ± 27.98	515 ± 69.991	0.048 ± 0.0054	745 ± 25.097	2.19 ± 0.684
Cold	152 ± 64.35	562 ± 164.32	0.103 ± 0.0167	715 ± 62.092	1.85 ± 1.914
Significance	NS	NS	NS	***	NS
Effect of feeding type					
Grazers	310 ± 57.92	506 ± 107.95	0.050 ± 0.0084	711 ± 38.1	2.34 ± 2.112
Intermediate feeders	209 ± 67.14	413 ± 113.16	0.045 ± 0.0084	723 ± 104.0	0.65 ± 9.708
Significance	NS	NS	***	NS	NS
Effect of ruminant type					
Buffalo	140 ± 163.43	519 ± 275.95	0.050 ± 0.0175	649 ± 95.64	3.33 ± 2.80
Cattle	140 ± 65.50	520 ± 124.65	0.052 ± 0.0093	726 ± 43.36	1.86 ± 2.487
Goats	307 ±75.71	414 ± 137.50	0.046 ± 0.0098	722 ± 105.98	0.65 ± 9.789
Sheep	218 ± 68.19	492 ± 133.48	0.048 ± 0.0098	703 ± 48.64	2.67 ± 2.472
Significance	NS	NS	***	NS	*

a, soluble fraction; b, slowly degradable fraction; PD, potential degradability and c, rate of degradation; Af, tropical rain forest; Aw, tropical savannah climate; BWh, hot arid desert climate; BSh, hot arid steppe climate; BSk, cold arid steppe climate; Csa, dry temperate climate with hot summers; Csb, dry temperate climate with warm summers; Cwa, dry winter temperate climate with hot summer; Cwb, dry winter temperate climate with warm summers; Cfa, hot summer temperate climate without dry season; Cfb, warm summer temperate climate without dry season; Dsb, cold dry climate with warm summers; Dfa, cold climate with hot summers and no dry season; Dfb, cold climate with warm summers and no dry season; Dfc, cold climate with warm summers and no dry season. * *p* < 0.05; ** *p* < 0.01; *** *p* < 0.001; NS, not significant (*p* > 0.05).

**Table 4 animals-11-00172-t004:** Covariate effects of feed sample and diet properties, ambient temperature and climatic region on rumen degradation of feeds.

Covariate Fixed Effects	a (g/kg)		b (g/kg)		c (per h)		PD (g/kg)		Lag (h)	
	*F*	*p*	*F*	*p*	*F*	*p*	*F*	*p*	*F*	*p*
Diet										
Browse	0.02	NS	0.20	NS	0.95	NS	1.50	NS	-	-
Silage	7.20	**	0.56	NS	0.20	NS	2.08	NS	5.68	*
CP (g/kg)	0.01	NS	1.05	NS	10.6	**	2.12	NS	0.07	NS
PFCC (g/kg)	0.23	NS	2.98	*	13.6	***	0.00	NS	0.89	NS
Feed sample										
DM (g/kg)	3.61	*	19.4	***	0.31	NS	1.46	NS	10.27	**
CP (g/kg)	27.3	***	311	***	6834	***	47.1	***	26.46	***
Ash (g/kg)	0.03	NS	60.1	***	114	***	27.3	***	0.04	NS
Silage	1.66	NS	0.05	NS	0.11	NS	0.98	NS	6.63	**
UT	0.00	NS	0.06	NS	5.18	*	0.02	NS	0.19	NS
PS (mm)	1.22	NS	0.47	NS	0.33	NS	0.01	NS	NS	0.08
Environment										
AT (°C)	0.02	NS	3.55	*	0.89	NS	8.09	***	1.31	NS
GR or IN	0.01	NS	0.00	NS	4.11	*	0.72	NS	-	-
Random effects										
S × I		**		**		***		***		NS

a, soluble fraction; b, slowly degradable fraction; c, rate of degradation; PD, potential degradability; S×I, Study×incubation time interactions; DM, dry matter; CP, crude protein; PFCC, protein-free cell contents; GR, grazing; IN, indoors; UT, urea treatment; PS, particle size. *P*, *p*-value; *F*, F-statistic; * *p* < 0.05; ** *p* < 0.01; *** *p* < 0.001; NS, not significant (*p* > 0.05).

**Table 5 animals-11-00172-t005:** Equations for linear regression between ambient temperature and dietary crude protein (independent variables) and slowly degradable fraction of fibre and potential degradability in the rumen.

Independent			Parameter Estimates					R^2^
Variables	Feed Type	N	Intercept	*p* _intercept_	Slope	*p* _slope_	RMSE	
**Slowly Degradable Fraction (g/kg)**								
AT (°C)	R	806	507.9 ± 13.62	<0.0001	−0.87 ^a^ ± 0.66	0.1890	144.2	0.002
	C	102	685.0 ± 24.23	<0.0001	−11.4 ^c^ ± 1.52	<0.0001	143.8	0.359
	MD	39	889.8 ± 34.47	<0.0001	−22.7 ^d^ ± 2.94	<0.0001	59.27	0.616
	All feeds	947	564.8 ± 11.71	<0.0001	−3.45 ^b^ ± 0.589	<0.0001	147.4	0.035
**Potential Degradability (g/kg)**								
AT (°C)	R	830	771.2 ± 12.91	<0.0001	−3.90 ^a^ ± 0.629	<0.0001	140.0	0.044
	C	105	876.7 ± 28.73	<0.0001	−7.60 ^b^ ± 1.83	<0.0001	173.2	0.144
	MD	39	1104 ± 31.27	<0.0001	−24.1 ^c^ ± 2.67	<0.0001	53.76	0.688
	All feeds	974	810.2 ± 11.24	<0.0001	−5.48 ^b^ ± 0.568	<0.0001	144.3	0.087
DCP (g/kg)	R	821	672.9 ± 14.76	<0.0001	0.23 ^c^ ± 0.117	0.0523	140.8	0.005
	C	105	800.1 ± 88.08	<0.0001	−0.16 ^b^ ± 0.687	0.8175	187.2	0.000
	MD	39	525.4 ± 26.25	<0.0001	1.52 ^a^ ± 0.125	<0.0001	43.07	0.799
	All feeds	965	653.6 ± 13.98	<0.0001	0.49 ^b^ ± 0.107	<0.0001	147.6	0.022
**Rate of Degradation (per h)**								
AT (°C)	R	847	0.05 ± 0.0031	<0.0001	−0.001 ^a^ ± 0.0002	0.0006	0.034	0.014
	C	108	0.07 ± 0.0086	<0.0001	0.002 ^a^ ± 0.0006	0.7093	0.053	0.003
	MD	39	0.06 ± 0.0086	<0.0001	−0.0005 ^a^ ± 0.0007	0.461	0.015	0.015
	All feeds	994	0.06 ± 0.0028	<0.0001	−0.0006 ^a^ ± 0.0001	<0.0001	0.037	0.019
AT (°C)	TLM	363	0.08 ± 0.0153	<0.0001	−0.001 ^a^ ± 0.00095	NS	0.134	0.060
	NTLM	634	0.06 ± 0.0035	<0.0001	−0.001 ^a^ ± 0.00016	<0.0001	0.033	0.035

^a,b,c,d^ Means in a column with different superscripts are significantly different (*p* < 0.05).DCP, dietary crude protein; AT, ambient temperature; N, number of data used; NTLM, no time-lag model; TLM, time lag model; R, roughages; C, concentrates; MD, mixed diets.

**Table 6 animals-11-00172-t006:** Equations for linear regression between chemical composition of feeds degraded in the rumen and ambient temperature.

Independent			Parameter Estimates					R^2^
Variables	Feed Type	N	Intercept	*p* _intercept_	Slope	*p* _slope_	RMSE	
**Dry Matter (g/kg DM)**								
AT (°C)	R	866	632.8 ± 26.829	<0.0001	4.04 ^b^ ± 1.321	0.0023	301.1	0.011
	C	109	896.3 ± 15.045	<0.0001	−1.04 ^c^ ± 0.972	0.287	92.88	0.011
	MD	39	−458.6 ± 130.3	0.0012	69.7 ^a^ ± 11.11	<0.0001	223.9	0.515
	All feeds	1014	653.1 ± 22.82	<0.0001	3.35 ^b^ ± 1.165	0.0042	301.8	0.008
**Neutral Detergent Fibre (g/kg DM)**								
AT (°C)	R	860	597.3 ± 15.73	<0.0001	−0.36 ^c^ ± 0.775	0.642	176.5	0.029
	C	107	173.9 ± 23.71	<0.0001	11.5 ^a^ ± 1.554	<0.0001	146.1	0.341
	MD	39	394.8 ± 38.97	<0.0001	10.5 ^a^ ± 3.325	0.0031	67.0	0.213
	All feeds	1006	486.6 ± 14.44	<0.0001	4.03 ^b^ ± 0.739	<0.0001	191	0.029
**Crude Protein Content (g/kg DM)**								
AT (°C)	R	860	107.2 ± 5.654	<0.0001	−0.014 ^b^ ± 0.28	0.9593	75.32	0.000
	C	109	148.6 ± 17.91	<0.0001	2.51 ^a^ ± 1.157	0.0325	110.6	0.042
	MD	39	342.1 ± 14.18	<0.0001	−11.8 ^d^ ± 1.210	<0.0001	24.39	0.720
	All feeds	1008	132.2 ± 5.695	<0.0001	−0.76 ^c^ ± 0.291	0.0094	75.32	0.007

^a,b,c,d^ Means in a column with different superscripts are significantly different (*p* < 0.05). AT, ambient temperature; R, roughages; C, concentrates; MD, mixed diets; N, number of data used.

**Table 7 animals-11-00172-t007:** Effects of feed sample and diet properties, ambient temperature, ruminant type and feeding type, and climatic region on rumen degradation of feeds.

Test of Fixed Effects	Lag Time Model		No Lag Time Model
Effect of Feed Type	c (per h)	Lag (h)	c (per h)
Roughages	0.048 ± 0.0098	2.773 ± 4.3112	0.045 ± 0.0182
Concentrates	0.079 ± 0.0101	0.960 ± 4.4111	-
Mixed diets	0.068 ± 0.1997	-	0.051 ± 0.0235
Significance	NS	***	NS
Effect of climatic region			
Tropical climates			
Af	-	6.067 ± 6.0983	0.029 ± 0.0304
Aw	0.032 ± 0.0700	1.540 ± 5.5654	0.040 ± 0.0197
Arid climates			
BSh	0.041 ± 0.0722	-	0.037 ± 0.0203
BSk	0.089 ± 0.0687	5.134 ± 5.5930	0.052 ± 0.0284
BWh	0.054 ± 0.0771	-	0.051 ± 0.0527
Temperate climates			
Cfa	0.054 ± 0.0674	0.619 ± 5.2530	0.045 ± 0.0220
Cfb	0.045 ± 0.0677	2.791 ± 6.0284	0.059 ± 0.0226
Csa	0.059 ± 0.0689	2.599 ± 5.7277	0.054 ± 0.0237
Csb	-	-	0.056 ± 0.0346
Cwa	0.057 ± 0.0684	0.579 ± 5.8280	0.127 ± 0.2257
Cwb	0.037 ± 0.0693	2.741 ± 5.5972	0.034 ± 0.0162
Cold climates			
Dfa	0.054 ± 0.0701	1.313 ± 5.9558	0.104 ± 0.0434
Dfb	0.234 ± 0.0805	2.033 ± 5.8402	-
Dfc	-	-	0.027 ± 0.0521
Dsb	-	-	0.032 ± 0.0313
Significance	NS	***	NS
Effect of climate type			
Tropical	0.032 ± 0.0145	2.626 ± 1.7446	0.039 ± 0.0133
Arid	0.073 ± 0.0137	3.137 ± 1.7201	0.044 ± 0.0154
Temperate	0.046 ± 0.0043	2.194 ± 0.5132	0.051 ± 0.0079
Cold	0.184 ± 0.0158	1.853 ± 1.9538	0.045 ± 0.0234
Significance	NS	NS	NS
Effect of feeding type			
Grazers	0.059 ± 0.0668	2.342 ± 2.112	0.045 ± 0.0195
Intermediate feeders	0.031 ± 0.0678	0.654 ± 9.708	0.064 ± 0.0195
Significance	NS	NS	***
Effect of ruminant type			
Buffalo	0.111 ± 0.0715	3.331 ± 2.7980	0.033 ± 0.0261
Cattle	0.058 ± 0.0681	1.865 ± 2.4869	0.049 ± 0.0203
Goats	0.031 ± 0.0699	0.654 ± 9.7886	0.064 ± 0.0202
Sheep	0.058 ± 0.0680	2.672 ± 2.4723	0.042 ± 0.0201
Significance	NS	*	***

Af, tropical rain forest; Aw, tropical savannah climate; BWh, hot arid desert climate; BSh, hot arid steppe climate; BSk, cold arid steppe climate; Csa, dry temperate climate with hot summers; Csb, dry temperate climate with warm summers; Cwa, dry winter temperate climate with hot summer; Cwb, dry winter temperate climate with warm summers; Cfa, hot summer temperate climate without dry season; Cfb, warm summer temperate climate without dry season; Dsb, cold dry climate with warm summers; Dfa, cold climate with hot summers and no dry season; Dfb, cold climate with warm summers and no dry season; Dfc, cold climate with warm summers and no dry season. * *p* < 0.05; *** *p* < 0.001; NS, not significant (*p* > 0.05).

**Table 8 animals-11-00172-t008:** Effects of feed type, climatic region and ambient temperature on chemical composition of feed samples incubated in the rumen.

Test of Fixed Effects	Chemical Composition Estimates (g/kg DM) (Mean ± SE)					
Effect of Feed Type	DM	CP	NDF	ADF	HEM	Ash
Roughages	822.5 ± 31.30	109.6 ± 8.42	606.5 ± 21.19	395 ± 14.23	211 ± 11.94	90.5 ± 4.58
Concentrates	878.8 ± 38.39	166.7 ± 11.04	355.7 ± 26.32	209 ± 18.15	147 ± 15.06	68.0 ± 5.88
Mixed diets	803.0 ± 63.28	131.9 ± 19.42	590.7 ± 43.67	367 ± 30.96	222 ± 25.39	95.1 ± 10.16
Significance	NS	***	***	***	***	***
Effect of climatic region						
Tropical climates						
Af	806 ± 159.02	83.3 ± 42.56	609 ± 107.82	365 ± 72.17	241 ± 60.67	122.9 ± 23.20
Aw	565 ± 70.91	133.2 ± 19.50	466.0 ± 48.31	297 ± 32.68	168 ± 27.35	88.5 ± 10.54
Arid climates						
BSh	895.5 ± 93.97	112.5 ± 25.69	560.1 ± 63.86	406 ± 43.11	154 ± 36.11	73.4 ± 13.91
BSk	888.4 ± 84.04	85.8 ± 23.87	660.9 ± 57.39	416 ± 39.39	232 ± 32.76	95.9 ± 12.78
BWh	865 ± 129.37	145.1 ± 35.01	481.2 ± 87.81	374 ± 59.04	107 ± 49.53	61.8 ± 19.01
Temperate climates						
Cfa	669.9 ± 72.04	188.1 ± 19.88	517.5 ± 48.96	310 ± 33.19	208 ± 27.75	99.3 ± 10.73
Cfb	781.3 ± 66.44	136.0 ± 18.27	491.6 ± 45.10	300 ± 30.55	193 ± 25.55	68.9 ± 9.87
Csa	772.2 ± 86.29	135.6 ± 23.56	419.7 ± 58.62	275 ± 39.56	144 ± 33.14	91.9 ± 12.76
Csb	493 ± 131.79	153.0 ± 36.01	400.4 ± 90.22	222 ± 60.67	177 ± 50.91	101.7 ± 19.55
Cwa	921 ± 91.90	123.7 ± 24.97	531.0 ± 62.41	315 ± 42.02	215 ± 35.23	70.8 ± 13.54
Cwb	835.3 ± 62.35	126.3 ± 17.13	509.6 ± 42.48	307 ± 28.73	206 ± 24.04	85.4 ± 9.26
Cold climates						
Dfa	884 ± 137.52	201.4 ± 37.25	420.8 ± 93.28	247 ± 62.76	176 ± 52.64	72.4 ± 20.22
Dfb	913 ± 137.48	187.5 ± 37.09	507.0 ± 93.19	273 ± 62.60	235 ± 52.54	48.9 ± 20.16
Dfc	890 ± 232.82	89.7 ± 62.30	657.0 ± 157.7	428 ± 105.6	231 ± 88.77	109.6 ± 33.96
Dsb	928 ± 158.12	139.8 ± 42.50	533 ± 107.19	321 ± 71.88	214 ± 60.38	76.7 ± 23.13
Significance	**	NS	NS	NS	NS	NS
Effect of covariates						
Feed sample						
Silage	***	NS	*	NS	***	*
Urea treatment	NS	NS	NS	*	*	NS
Environmental factors						
Ambient temperature	*	NS	NS	NS	NS	NS
Test of random effects						
Study×incubation time	***	***	***	***	***	***

DM, dry matter; CP, crude protein, NDF, neutral detergent fibre; ADF, acid detergent fibre; HEM, hemicellulose; Af, tropical rain forest; Aw, tropical savannah climate; BWh, hot arid desert climate; BSh, hot arid steppe climate; BSk, cold arid steppe climate; Csa, dry temperate climate with hot summers; Csb, dry temperate climate with warm summers; Cwa, dry winter temperate climate with hot summer; Cwb, dry winter temperate climate with warm summers; Cfa, hot summer temperate climate without dry season; Cfb, warm summer temperate climate without dry season; Dsb, cold dry climate with warm summers; Dfa, cold climate with hot summers and no dry season; Dfb, cold climate with warm summers and no dry season; Dfc, cold climate with warm summers and no dry season. * *p* < 0.05; ** *p* < 0.01; *** *p* < 0.001; NS, not significant (*p* > 0.05).

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
