# Peer review of "Consequences of Increases in Ambient Temperature and Effect of Climate Type on Digestibility of Forages by Ruminants: A Meta-Analysis in Relation to Global Warming"

_animals, 2021, doi:10.3390/ani11010172_

Round 1

Reviewer 1 Report

Dear Authors, the paper 'Consequences of increases in ambient temperature and effect of climate type on digestibility of forages by ruminants: a meta-analysis in relation to global warming" is an interesting article with many up-to date findings. 

The summary, abstract and keywords are a good represetative of the text. 

The introduction provides a good bacground on the field, however I would suggest to update the reference list, to highlight the importance of carried discussion. Moreover, the Authors should mention the main aim and then show how these was met in the concluding section. It would be more concise.

line 63 - why is it unclear? are there really no information in the literature? doi:10.1017/S0954422408960674; doi.org/10.2527/jas1978.4661768x

the clue for dataset choice needs to be clearly explained, it is very easy to manipulate the results, if certain datasets are chosen, please explain.

How were the feed sample properties affecting degradability identified? Usually a statistical plan, i.e. Plackett-Burmann or literature reports should be applied.

Please explain randomly the main findings of Harris et al, as you adopted them for your data.

The p-value of 25 is very high, please explain why was it set at this level?

What is the physical sense of strong positive correlation, i.e. between ADF or HEM and NDF? The discussion of table 2 is rather poor. Please mention if the data were normalized?

As you have presented very much results and their processing, maybe an anova dagfam would be of good value, between the model-predicted and the obtained results, please consider.

The results are genetarally supporting the conclusions.

The conclusions are concise, maybe add a few bullet points regarding the contribution of yoour approach to the field

Reviewer 2 Report

  1. Expand the abstract with a couple of specific values for effect of cold stress.
  2. Lines 85-88 - How did you account for things like pectin, beta-glucans, fat and minerals and vitamins in the formula in line 86. Need to clarify this. Does this formula overestimate sugars and starches?
  3. Lines 93-94 - You list a number of feed properties (ADF, hemicellulose, ash). How were they determined. How does this fit with the formula in line 86. Same comment on lines 102-103.
  4. Line 157 - How many mixed diets were in the dataset? Any index of the ratio of forage to grain in these samples? 1
  5. Lines 358-361 - Good comments. Same fo lines 366-368.
  6. Figure 1 is very helpful.
  7. Conclusions - Need to expand to add results for cold stress
